# OpenReview forum: "Persistent Homology Captures the Generalization of Neural Networks Without A Validation Set"
_ICLR.cc/2022/Conference — ICLR 2022 Submitted_

### Official Review · Reviewer_JhC7 · 2021-10-28

**Correctness:** 1
**Technical Novelty And Significance:** 3
**Empirical Novelty And Significance:** 2
**Recommendation:** 3
**Confidence:** 4

**Main Review:**

# Strengths and Weaknesses

## Strengths:
- The paper is well written overall (aside for few typos, see Minor comments).
- Experimental report is extensive given the amount of complementary plots in the supplementary material.
- The methodology proposed by the paper is original and interesting. The construction of topological descriptor for networks has interesting properties, theoretically improving for instance on the approach proposed by Rieck et al. (at the price of computational efficiency though).

## Weaknesses :

- My main concern with the paper is that its conclusions seem too optimistic. From my understanding, saying that the "*proposed measure strongly correlate with validation accuracy*" (Section 7) and that "*the generalization error of a neural network could be intrinsically estimated without any holdout set*" (abstract) would mean for it to be useful that :
  - (a) If the network starts overfitting, that is the validation accuracy starts decreasing (while the training accuracy presumably keeps increasing), this should be reflected in the curve of (cumulative) distances between (vectorization of) PDs. On Figure 7, b, for instance, there is---as far as I understand---no way to infer from the line-curve (PD distances) that the network is overfitting.
  - (b) To be able to predict the generalization gap, one would expect a change of behavior when the network properly learns (e.g. MNIST) vs when it does not (e.g. CIFAR-10-MLP) ; (here, I assume that the training accuracy is going high---it as not been reported from what I can tell so we cannot be sure what the generalization gap is. Hence I assume that low validation accuracy $\Leftrightarrow$ high gap.). From Figure 8.a and 8.b for instance, how is one supposed to infer that one network is learning something useful while the other does not?

I am worried that what is interpreted as a strong correlation may only be a consequence of the networks parameters optimization/convergence. During optimization, as the step size goes to zero using RMSprop, it is not unlikely that the network weights variations (hence successive diagram distances) are getting smaller, yielding this concave-shape for the plots of cumulative diagram distances and for the *training* accuracy---thus for the validation accuracy when the network learns properly (i.e. validation accuracy $\simeq$ training accuracy).

Furthermore, when looking at plots in the supplementary material (for instance Figure 53), it seems that the reasonably correct correlation coefficients obtained are in great part due to the correlation *before* network parameters convergence (bottom-left quadrant: low topological cumulative, low validation score), which are not very interesting. On the top-right quadrant (when the network is likely to be "ready"), things get messier.

In the same vein, the sentence "*(...) validation accuracy depends on the specificity of the data sampled in the validation subset, while the homological convergence is independent of the validation data*" (end of section 6) is somewhat contradictory with the claim of the paper that homological properties of the network "*captures the generalization of neural networks without a validation set*".

All in all, and unless I have missed something crucial, I feel that the empirical results (which are quite numerous, thanks!) provided by this work are, unfortunately, not convincing.

- A second important concern is the intersection with the Anonymous concurrent work (available in the supplementary material). With few exceptions, everything is the same up to the end of section 5 : same construction of topological descriptors (flag complex, filtration by Eq (1)), same diagram vectorizations, same datasets, same changes of architecture (number of layers, layer widths)... So that, roughly, what differs in the two paper is the experimental measurement: this paper studies what happen at training time for a given network, while the concurrent one makes cross-networks comparisons. Though the two papers cover different objectives, it diminishes the intrinsic contributions of the current one, which are now limited to the experimental results which are, as said above, insufficiently convincing in my opinion.
- As acknowledged by the authors, this method does not scale and is only applicable to reasonable MLP; even though requiring high-end hardware and a lot of time.
- The paper is not supported by any theoretical result that would motivate the use of their approach.


# Minor comments
- (Typo) p2, give-->given
- (Typo) End of Section 3 : "They claim. However". It seems a sentence is missing.
- (Typo) Start of p5 : an smoothing --> a smoothing
- (Suggestion) Eq 1 : use \max and \left( \right).
- (Suggestion) Numbering cases with "1. blabla... 2. blabla" hinders readability. Perhaps using something in the vein of "*(i)* blabla, *(ii)* blabla" would be better.
- (Clarity) The transition between filtrations of simplicial complex and persistence diagrams may be a bit too short (I am referring to the single sentence "The sequence of homology groups is calculated by varying the parameter $\epsilon$ to obtain the persistence homology diagram.")
- (Typo) p5 : a reference is missing ("as in ?").
- end of p7, "the homology does not seem to converge". I do not understand this claim. If the network (parameters) converge, so does the persistence diagrams. If my understanding is correct, RMSprop should always yield convergence (it uses a decay factor for the step size); so I guess more epochs are needed.
- p9 : "our approach computes the exact PDs distance", I think the exact PDs are computed in first place (not the distances).
- Results in Table 1 for Heat and Silhouette are the exact same. Is it an unfortunate copy-paste?
- Supplementary material, Figures 42-53 : how can it be that the topological cumulative decreases (given that the validation score seems monotonic, I guess that the axes labels have been switched?)? Also, why does the validation score reaches 1, while it is supposed to be much lower on some model? (I guess a normalization has been applied).
- The paper claims that "*in CNNs, the correlation are (...) still usually above 0.8*", but unless I misread the tables 5,6, 9, and 10, this seems to be exaggerated. CIFAR100CNN + number of layers $\geq 4$ is one the only instance where this seems to hold.

**Summary Of The Paper:**

This paper experimentally investigates the (cor)relation between the validation accuracy achieved by a neural network and a variation of a topological descriptor built on top of the network: the persistence diagram of a directed flag complex built on top of the network.

The main claim is that there is a high correlation between the validation accuracy and this "topological distance", so that the later may be used when a validation set is not available.

**Summary Of The Review:**

Though I appreciate the ambitions of the work, I feel that the empirical results are, unfortunately, not convincing. The absence of theoretical results and the very strong similarities with the concurrent submitted paper diminishes it's impact as well.

---

> ### Author Response · Authors · 2021-11-21
> **Anwser to Official Review of Paper4090 by Reviewer Fo4q**
>
> >(a) If the network starts overfitting, that is the validation accuracy starts decreasing ...
>
> Figure 7b shows precisely the control experiments where the described process does not take place. These are the 5 control experiments in which the order of the training examples and the initialization of each neural network is different. As can be seen, the distances of the PDs remain very similar in their evolution.
>
> >(b) To be able to predict the generalization gap,...
>
> Note the detail of the right scale (Validation score) in each of the figures. In one case the values range between 0.4-1 and in the second case between 0.1-0.5.
> The error curve is always much lower than the validation curve since it is the value to be minimized. The article is focused on validation analysis and the diagrams were not clear if the number of curves was doubled to include the error. We could include additional diagrams in the supplementary material.
>
> >I am worried that what is interpreted as a strong correlation may only be a consequence of the networks parameters >optimization/convergence. ...
>
> Indeed, when the modification in the weights of the neural network is very small, the variations in the topology can be small. Topologically this is not so evident given that connections and disconnections of nodes can occur with an impact on the homology groups and therefore on the calculation of the persistent homology. This observation would apply only to the final part of the curves (the last training epochs).
>
> However, we believe that the informative part of these graphs occurs in the first epochs (<10). Notice how in some cases the topological convergence is very fast and stable afterward. That is, the process of topology learning and numerical convergence are not synchronized. The persistent homology information anticipates the subsequent behavior of the validation curve.
>
> >Furthermore, when looking at plots in the supplementary material (for instance Figure 53),...
>
> This is precisely one of the points we want to emphasize in the paper. When we perform the control experiments, the correlation between the different initializations and reordering of examples is given from the beginning. However, when we experiment with parameters that alter the learning process (dropout, learning rate, layer sizes, number of layers,...) the convergence in the validation and in the distances of the PDs are very different. What we argue is that it is possible to draw conclusions about the generalization of the NN being trained without information from the validation dataset and in an anticipatory manner.
>
> >In the same vein, the sentence "(...) validation accuracy depends on the specificity of the data sampled ...
>
> I do not understand the contradiction indicated. The curves of the evolution of the distance between PDs is calculated from training data (during NN training) while the validation curves use a different data set.
>
> >A second important concern is the intersection with the Anonymous concurrent work ...
>
> Both papers use similar representations and metrics but are applied to two different problems. On the one hand, the similarity between NNs trained for different problems is analyzed (cross-networks comparisons). On the other hand, the learning process is analyzed, in particular, the generalization of the NN.
>
> >As acknowledged by the authors, this method does not scale and is only applicable to reasonable MLP; even though requiring high-end >hardware and a lot of time.
>
> The experiments performed are computationally expensive due to the large number of combinations of experiments performed (dropout, learning rate, architectures, problems, weight initialization, training example errors, ...).
> There are many ways to speed up the proposed computations (precision of the persistent homology parameter, vectorization dimension, etc.). In this paper, we have tried to carry out the study with as much detailed precision as possible.
>
> >The paper is not supported by any theoretical result that would motivate the use of their approach.
>
> The theoretical justification that could support this experimental paper would be a completely new theory of function approximation. This theory is not available and was not presented in this publication. Our paper is only an experimental study.
> We analyze how a numerical convergence, governed by the backpropagation method, is related to a topological convergence associated with the function but showing distinct properties of the convergence process, in our point of view, it is a parallel convergence process with interesting properties: the anticipation of the validation accuracy using only training data. We understand that this is a relevant result for the study of generalization in NNs.
>
> >Minor comments
> Ok. We will review them

---

### Official Review · Reviewer_Fo4q · 2021-11-01

**Correctness:** 2
**Technical Novelty And Significance:** 3
**Empirical Novelty And Significance:** 2
**Recommendation:** 3
**Confidence:** 5

**Details Of Ethics Concerns:**

Authors were bordering on passive--aggressive and behaved in an abrasive manner with all reviewers. Instead of updating/revising the paper, much was spent on nit-picking irrelevant details. Overall, I have the feeling that the hours I put into reviewing this paper and engaging with the authors were rather wasted. I wish the authors would take some of the comments to heart—my summary with a link to a recent NeurIPS paper demonstrates how a successful experimental setup could look like; I honestly don't understand why the authors appear to be so adamant and hostile in these exchanges.

**Main Review:**

I enjoyed the ideas presented in this paper; the analysis of
generalisation performance is a highly relevant and timely, and indeed
constitutes one of the largest obstacle toward employing machine
learning techniques in the wild. Understanding generalisation without
requiring additional data has the potential to improve machine learning
techniques to a substantial extent.

That being said, the current paper suffers from some issues, which
prevent my endorsement at this point:

1. Clarity: while background information on topological data analysis is
   provided (which I appreciate; in particular in light of the fact that
   TDA is still a rather novel occurrence at ML conferences), the method
   itself could be described in more detail. In particular, some aspects
   should be discussed in Section 4:

    - The use of additional representations for topological features
      needs to be clarified. At present, the main paper does not contain
      any description of the challenges in calculating distances or the
      need for additional representations. At the very least, a brief
      description of these topological representations is required, so
      that readers may better understand the remainder of the paper.
      I find it particularly problematic that no stability guarantees of
      these representations are discussed; they are not *just* drop-in
      replacements for the bottleneck or Wasserstein distances.

    - Overall, this section should be rewritten with a clear 'roadmap'
      in mind. What is the problem you want to tackle and how do you
      intend to tackle it in this paper?

2. Delineation to existing work: The ICLR paper by Rieck et al. (2019b)
   appears to already contain a large amount of the material proposed in
   this paper. A cursory reading shows that Rieck et al. even discuss
   generalisation performance in an early stopping setting (more about
   this later). It is therefore critical that the contributions of this
   paper are more clearly delineated from existing work. From my
   understanding of both papers, I would say that the current submission
   improves on the following aspects:

    - Choice of filtration for such neural networks

    - The use of other topological representations (whereas Rieck et al.
      only use a summary statistic of persistence diagrams).

   These differences (and potentially all others that I missed) should be
   briefly discussed; the advantage would be that the paper can refer to
   the previous publication as a justification of the method itself!

   The previous work by Rieck et al. also proposes a set of experiments
   that would be very fitting to perform here: the experiments proposed
   in this paper could be substantially strengthened by the addition of certain
   (simple) monitoring baselines that serve to showcase the added value
   of employing topological information in the first place here. Rieck et
   al. compared their topological measure to a 'patience criterion' based
   on validation loss, thus assessing the difference in performance
   and epochs after the early stopping criterion was applied). In my
   opinion, this is most important experiment here when assessing
   validation performance. Also, it would be interesting to add simple
   baselines measures to check that these are *insufficient* to be used
   alone. For instance, a non-topological baseline that could be assessed
   in this context would be a distribution of weights; it should not be
   possible to assess validation performance with this baseline.

2. Experimental depth: the experimental section is lacking depth, in
   particular given the specific goals of this paper. The depicted plots
   and correlations are useful, but don't serve to highlight the
   benefits of the proposed method. I would therefore suggest a more
   thorough experimental setup such as the one shown in the supplement
   of Rieck et al. (2019b): as mentioned in the previous point, their
   proposed measure of network complexity is shown as additional early
   stopping criterion (without an hold-out validation data set as well).

   A post-hoc experiment of this sort would be extremely useful in
   demonstrating the utility of the method, and it would even enable the
   quantification of certain measures. The correlation is also useful in
   this context, but since the goal is to assess the generalisation
   error, an experiment in which the new measure is directly applied
   would be extremely worthwhile.

Please see below for detailed comments.

## Detailed comments

- The first contribution reads a little bit like a sentence fragment to
  me. I would suggest to use the terms 'silhouette distance' and 'heat
  kernel distance' instead of plain 'silhouette' and 'heat'.

- The terminology 'homological convergence' is slightly misleading
  because the paper studies persistent homology, as far as I understand.

- The introduction of topological concepts in the background section
  jumps between the 'geometrical' and the 'abstract'; I understand that
  both perspectives are valid, but I would suggest to choose only
  a single one.

- The discussion of chains and boundaries requires more explanations (or
  could be partially relegated to the supplementary materials). For
  instance, please clarify the 'signed combination.' However,
  definitions need to appear in the main text, not only in the
  supplementary materials.

- In figure 4, the $3$-simplex should consist of four $2$-simplices, but
  only two are shown. Either clarify this in the captions or update the
  figure accordingly.

- The introduction of filtrations is not motivated; the 'nested family
  of simplicial complexes' could be introduced by building more
  intuition here.

- The related work discussion should, as mentioned above, delineate this
  paper from Rieck et al. (2019b).

- The heading 'Algebraic Topology object' is rather confusing to me;
  I would suggest to rewrite it as 'directed flag complex', for
  instance.

- As outlined above, the distance calculation lacks information about
  stability and other guarantees. There is also a broken reference on p.
  5, which needs to be rectified.

- In the experimental section, the use of the MLP should be explained in
  more detail. What does it do and how does it work exactly?

- I do not understand the 'input order' experiment; does it refer to
  ordering of samples (i.e. on the batch level)?

- The use of additional topological descriptors makes the use of the
  word 'distance' imprecise: the bottleneck and Wasserstein
  distances are metrics in the mathematical sense; while there *is*
  a well-defined metric between, for instance, persistence landscapes or
  silhouettes, this is only (to a certain extent) an approximation to
  the bottleneck/Wasserstein metrics. This distinction should be made
  clearer in the paper.

- Figure 6 should add standard deviations of the curves. An additional
  clarification of the content of this figure would also be appreciated
  by readers; I found the discussion to be slightly confusing because
  I did not get the relevance of the cumulative distance. The statement
  'the evolution of the homological convergence [...] seems to be very
  similar to the one of the validation score' also needs some
  clarification. Except for Figure 6b, where I observe some oscillation
  behaviour and maybe the tendency to converge, I don't observe
  convergence anywhere else.

- The same applies to Figures 7–9. Adding some more details here will
  result in a clearer description of the method. Maybe some of the
  curves could also be relegated to the supplementary materials?

- I would also suggest to investigate the use of energy distances or
  energy correlations, as these measures are more flexible and capable
  of assessing more than just linear dependencies between two
  variables. See [*Energy statistics: A class of statistics based on distances*](https://www.sciencedirect.com/science/article/abs/pii/S0378375813000633)
  for more details.

## Terminology and style

For the most part, the paper is written well. I have some suggestions
concerning the style:

- Please use `\citep` and `\citet` consistently when using `natbib`. At
  present, citations are directly appearing in the text without any
  enclosing parentheses. For instance, in the related work section,
  the sentence 'for improving the training procedure of the models Hofer
  et al. (2020); [...]' should use `\citep` in order to obtain a proper
  parenthetical citation.

- can't --> cannot (likewise for other contractions; I admit that this
  is a personal preference, so feel free to ignore it)

- 'contained simplicial complex' --> 'nested simplicial complexes'

- The paper employs non-standard terminology in certain places. The
  diagram, for instance, is called 'persistence diagram', no
  'persistence homology diagram'. Please ensure consistency with
  existing papers here. The same applies to capitalisation; I personally
  see no need to write 'Persistence Diagram', but if this spelling is
  used, it should be used consistently.

- 'complex clique' --> Use the term 'clique complex'

- 'module $Z_i(K)$' --> 'modulo $Z_i(K)$' (the paper is discussing the
  quotient operation)

- 'non-cumulative homology': I think the word 'distances' is missing
  here.

- The use of 'Means mean' and 'Deviations mean' is slightly confusing;
  I would suggest to show follow the format $\mu \pm \sigma$, with $\mu$
  being the mean and $\sigma$ being some measure of variance, such as
  the standard deviation.

- For the references list, I would suggest to carefully check which
  version of a paper is being cited. Numerous papers mentioned in the
  bibliography have already been published as book chapters, papers,
  etc.

- The preprint by Guss and Salakhutdinov is cited twice.



**Summary Of The Paper:**

This paper analyses the training of neural networks from a topological
perspective, presenting a pipeline that can measure (pseudo) distances
between the network's weights during training. Such information is then
employed to study the generalisation error of a neural network.

In contrast to existing methods for estimating this error, this paper
does *not* require a specific hold-out data set, as topological features
of the neural network are monitored during training. This frees up
additional data for fitting, which can be highly relevant in the sparse
data regime.

**Summary Of The Review:**

While I am very excited about topology-based approaches that aim to
understand the training or testing behaviour of neural networks,
I cannot endorse this paper for publication yet.

The current write-up is suffering from several issues, which need to be
rectified in a **major revision** before reaching the quality standards
of ICLR. These issues include:

1. Lack of clarity: an improved introduction to topological concepts is
   required and the proposed method needs to be compared more with
   existing topology-based approaches (in particular since it does not
   meet the requirements of a metric in the mathematical sense, an
   analysis of approximation guarantees is crucial).

2. Lack of experimental depth and delineation to existing work: since
   the express goal of this paper is to analyse generalisation
   properties of neural networks based on their topological properties,
   a comparison with existing work (Rieck et al., 2019b) and an improved
   experimental suite (containing previous work *and* non-topological
   baselines) is critical for corroborating the claims of this paper.

**Updated after rebuttal**: A lot has been discussed during this rebuttal period. I would strongly
recommend to pick up some of the suggestions of reviewers in order to improve the paper. A recent paper
by [Birdal et al.](https://papers.nips.cc/paper/2021/hash/35a12c43227f217207d4e06ffefe39d3-Abstract.html)
demonstrates how to successfully assess generalisation performance (disclaimer: I am *not* one of the authors
of said paper). I hope that this may serve as a partial inspiration.

---

> ### Author Response · Authors · 2021-11-21
> **Anwser to Official Review of Paper4090 by Reviewer Fo4q**
>
> >(a) If the network starts overfitting, that is the validation accuracy starts decreasing ...
> Figure 7b shows precisely the control experiments where the described process does not take place. These are the 5 control experiments in which the order of the training examples and the initialization of each neural network is different. As can be seen, the distances of the PDs remain very similar in their evolution.
>
> >(b) To be able to predict the generalization gap,...
> Note the detail of the right scale (Validation score) in each of the figures. In one case the values range between 0.4-1 and in the second case between 0.1-0.5.
> The error curve is always much lower than the validation curve since it is the value to be minimized. The article is focused on validation analysis and the diagrams were not clear if the number of curves were doubled to include the error. We could include additional diagrams in the supplementary material.
>
> >I am worried that what is interpreted as a strong correlation may only be a consequence of the networks parameters >optimization/convergence. ...
> Indeed, when the modification in the weights of the neural network is very small, the variations in the topology can be small. Topologically this is not so evident given that connections and disconnections of nodes can occur with an impact on the homology groups and therefore on the calculation of the persistent homology. This observation would apply only to the final part of the curves (the last training epochs).
>
> However, we believe that the informative part of these graphs occurs in the first epochs (<10). Notice how in some cases the topological convergence is very fast and stable afterwards. That is, the process of topology learning and numerical convergence are not synchronized. The persistent homology information anticipates the subsequent behavior of the validation curve.
>
> >Furthermore, when looking at plots in the supplementary material (for instance Figure 53),...
> This is precisely one of the points we want to emphasize in the paper. When we perform the control experiments, the correlation between the different initializations and reordering of examples is given from the beginning. However, when we experiment with parameters that alter the learning process (dropout, learning rate, layer sizes, number of layers,...) the convergence in the validation and in the distances of the PDs are very different. What we argue is that it is possible to draw conclusions about the generalization of the NN being trained without information from the validation dataset and in an anticipatory manner.
>
> >In the same vein, the sentence "(...) validation accuracy depends on the specificity of the data sampled ...
> I do not understand the contradiction indicated. The curves of the evolution of the distance between PDs is calculated from training data (during NN training) while the validation curves use a different data set.
>
> >A second important concern is the intersection with the Anonymous concurrent work ...
> Both papers use similar representations and metrics but applied to two different problems. On the one hand, the similarity between NNs trained for different problems is analyzed (cross-networks comparisons). On the other hand, the learning process is analyzed, in particular, the generalization of the NN.
>
> >As acknowledged by the authors, this method does not scale and is only applicable to reasonable MLP; even though requiring high-end >hardware and a lot of time.
> The experiments performed are computationally expensive due to the large number of combinations of experiments performed (dropout, learning rate, architectures, problems, weight initialization, training example errors, ...).
> There are many ways to speed up the proposed computations (precision of the persistent homology parameter, vectorization dimension, etc.). In this paper we have tried to carry out the study in as much detailed precision as possible.
>
> >The paper is not supported by any theoretical result that would motivate the use of their approach.
> The theoretical justification that could support this experimental paper would be a completely new theory of function approximation. This theory is not available and probably was not presented in this publication. Our paper is only an experimental study.
> We analyze how a numerical convergence, governed by the backpropagation method, is related to a topological convergence associated to the function but showing distinct properties of the convergence process, in our point of view, it is a parallel convergence process with interesting properties: the anticipation of the validation accuracy using only training data. We understand that this is a relevant result for the study of generalization in NNs.
>
> >Minor comments
> Ok. We will review them

---

> ### Author Response · Authors · 2021-11-22
> **Anwser to Official Review of Paper4090 by Reviewer Fo4q**
>
> We have found the following problems in existing topological models in the literature (mainly Rieck et al. and Corneanu et al.).
>
> 1) Invariance to superfluous cycle insertions
> Suppose that given an NN we perform the following transformation. We add two neurons and connect their input to a single neuron in a lower layer and their two outputs to a single neuron in an upper layer. If we assign opposite weights to the edges connecting these new neurons to the lower layer (e.g., a and -a) and similarly assign opposite weights to the edges connecting them to the upper layer (e.g., b and -b), we obtain another equivalent neural network. Since the weights are opposite, the net weight added on the upper layer neuron is zero and therefore both NNs generate the same outputs for any input.
>
> We expect a topological model representing the NN to be invariant to this kind of equivalence operation; the addition of superfluous cycles (I call them cycles because if we take sense of their edges according to their sign, as we do in our model, it is a homologically non-significant cycle, for any degree homology group).
>
> However, in Rieck et al.'s neural persistence model, since the a and b values can be as high or as low as we want, it is possible to cancel out, relatively speaking, the weights of the rest of the axes that are subsequently used in the neural persistence calculation.
>
>
> 2) Invariance to identity layer insertions
> Suppose we add an identity layer. That is, for any given NN layer, we duplicate its neurons and trivially connect them with edges whose weight is maximum.
>
> The neural network obtained after this transformation is functionally equivalent to the original NN. In our persistent homology model, as these edges have the maximum weight, the transformed value of the weight is zero, i.e. the duplicate nodes are always connected. From our point of view of persistent homology, no homology group of any degree is altered.
>
> From the layer pair point of view of Rieck neural persistence, four 0-homology elements are generated (since H_0 only evaluates connectivity), as many as the number of neurons in the duplicated layer exists.
>
>
> 3) Non-planar neural networks
> Sometimes we want to represent neural networks that are not planar, i.e. that connect several networks, for example by creating an assembly between them, or because they are networks that use multiple connections between non-consecutive layers (as in the example of residual connections). In these cases, the Rieck et al. model is not applicable because it is only valid for planar NNs with contiguous neuronal layers.
>
>
> 4) Models based on correlation between activations of neurons
> Finally, we have found models based on correlations of the neuron activations (Corneanu et al.). They form a undirected weighted graph that represents these correlations (undirected graph since it does not take into account that the correlation can be negative). The persistent homology analysis is then performed on this graph. We have not found this type of model is useful because they depend on the input data and not only on the function under study. We wanted to study neural networks as functions over the entire input domain, regardless of the data used for training.
>
>
>
> In general, we think that in order to make a complete homology representation of a NN we should use higher order homology groups (we have found important groups up to degree 3 but there could be cases where the complexity is higher). This is necessary to find non-trivial higher order homology structures that are specific and characterise the NN. We also see it as very important to employ directed graphs, and therefore to use directed flag homology.

---

> > ### Comment · Reviewer_Fo4q · 2021-11-22
> > **Fully agree**
> >
> > I fully agree with the points you are writing; extending the model by Rieck et al. is indeed very necessary in order to capture more complicated patterns in neural networks. These are not necessarily the concerns I raised, though—I think we all agree that there is a gap in the literature that can be filled here.

---

> ### Author Response · Authors · 2021-11-22
> **Reply to Review of Paper4090 by Reviewer Fo4q**
>
> Reviewing the two reviews (Neurips, ICRL), which at first sight seemed identical, I have found some new points that I will now answer:
>
> >The previous work by Rieck et al. also proposes a set of experiments that would be very fitting to perform here: the experiments proposed in this paper could be substantially strengthened by the addition of certain (simple) monitoring baselines that serve to showcase the added value of employing topological information in the first place here. Rieck et al. compared their topological measure to a 'patience criterion' based on validation loss, thus assessing the difference in performance and epochs after the early stopping criterion was applied). In my opinion, this is most important experiment here when assessing validation performance. Also, it would be interesting to add simple baselines measures to check that these are insufficient to be used alone. For instance, a non-topological baseline that could be assessed in this context would be a distribution of weights; it should not be possible to assess validation performance with this baseline.
>
> I think we have already answered the differences between representations and distances. If necessary we include it here.
>
> >In figure 4, the -simplex should consist of four -simplices, but only two are shown. Either clarify this in the captions or update the figure accordingly.
> Ok. We will review and clarify captions.
>
> >module ' --> 'modulo ' (the paper is discussing the quotient operation)
> Ok
>
> >non-cumulative homology': I think the word 'distances' is missing here.
> Ok
>
> >The preprint by Guss and Salakhutdinov is cited twice.
> Ok. We will remove this duplicated cite.

---

> > ### Author Response · Authors · 2021-11-22
> > **Anwser to Official Review of Paper4090 by Reviewer Fo4q**
> >
> > We have found the following problems in existing topological models in the literature (mainly Rieck et al. and Corneanu et al.).
> >
> > ## (1) Invariance to superfluous cycle insertions
> >
> > Suppose that given a NN we perform the following transformation: We add two neurons and connect their input to a single neuron in a lower layer and their two outputs to a single neuron in an upper layer. If we assign opposite weights to the edges connecting these new neurons to the lower layer (e.g., a and -a) and similarly assign opposite weights to the edges connecting them to the upper layer (e.g., b and -b), we obtain another equivalent neural network. Since the weights are opposite, the net weight added on the upper layer neuron is zero and therefore both NNs generate the same outputs for any input.
> >
> > We expect a topological model representing the NN to be invariant to this kind of equivalence operation; the addition of superfluous cycles (I call them cycles because if we take sense of their edges according to their sign, as we do in our model, it is a homologically non-significant cycle, for any degree homology group).
> >
> > However, in Rieck et al. neural persistence model, since the a and b values can be as high or as low as we want, it is possible to cancel them out, relatively speaking, the weights of the rest of the axes that are subsequently used in the neural persistence calculation.
> >
> > ## (2) Invariance to identity layer insertions
> > Suppose we add an identity layer. That is, for any given NN layer, we duplicate its neurons and trivially connect them with edges whose weight is maximum.
> >
> > The neural network obtained after this transformation is functionally equivalent to the original NN. In our persistent homology model, as these edges have the maximum weight, the transformed value of the weight is zero, i.e. the duplicate nodes are always connected. From our point of view of persistent homology, no homology group of any degree is altered.
> >
> > From the layer pair point of view of Rieck neural persistence, four 0-homology elements are generated (since H_0 only evaluates connectivity), as many as the number of neurons in the duplicated layer exists.
> >
> >
> > ## (3) Non-planar neural networks
> > Sometimes we want to represent neural networks that are not planar, i.e. that connect several networks (like siamese NNs), for example by creating an assembly between them, or because they are networks that use multiple connections between non-consecutive layers (as in the example of residual connections). In these cases, the Rieck et al. model is not applicable because it is only valid for planar NNs with contiguous layers.
> >
> >
> > ## (4) Models based on correlation between activations of neurons
> > Finally, we have found models based on correlations of the neuron activations (Corneanu et al.). They form an undirected weighted graph that represents these correlations (it is an undirected graph since it does not take into account that the correlation can be negative). The persistent homology analysis is then performed on this graph. We have not found this type of model useful as they depend on the input data and not only on the function under study. We wanted to study neural networks as functions over the entire input domain, regardless of the data used for training.
> >
> > We do not think data-based approach is positive for these reasons (among others):
> > 1. The need to pass data through the network. How much data?
> > 2. It is not the same to pass {training, validation, testing} data.
> > 3. The subset may not be representative.
> > 4. Topology/Structure of network is not taken into account. It is not being captured.
> > 5. Activations may not be enough.
> >
> > ## Final thoughts
> >
> > In general, we think that in order to make a complete homology representation of a NN we should use higher order homology groups (we have found important groups up to degree 3 but there could be cases where the complexity is higher). This is necessary to find non-trivial higher order homology structures that are specific and characterise the NN. We also see it as very important to employ directed graphs, and therefore to use directed flag homology.

---

> > ### Comment · Reviewer_Fo4q · 2021-11-22
> > **re: your review**
> >
> > > I think we have already answered the differences between representations and distances. If necessary we include it here.
> >
> > Can you please clarify what your intentions are wrt. such experiments, which would be useful in order to further characterise the generalisation performance?

---

### Official Review · Reviewer_eLVu · 2021-11-01

**Correctness:** 1
**Technical Novelty And Significance:** 1
**Empirical Novelty And Significance:** 2
**Recommendation:** 1
**Confidence:** 4

**Main Review:**

One merit of the paper is, to put it in very broad terms, the idea of
applying tools from topological data analysis to the problem of
understanding generalization in deep neural networks. In principle,
this is an interesting problem from a theoretical standpoint, and the
idea of applying TDA is timely in the sense that many fields are now
in the process of "discovering" topological techniques, and it is good
to see some exploration of what they can be useful for. Unfortunately
however, in my opinion the strengths of the paper stop here. I will
now go in detail over what I consider to be its major weaknesses.

Empirical studies can be valuable, but the complete absence of any
theoretical justification (or even just an intuitively plausible
story) for the main argument means that the experimental evidence
presented carries all the burden of convincing the reader. My main
concern with the paper is that, unfortunately, the analyses presented
are nowhere close to the standard of rigor and thoroughness one would
expect in this case. In particular:

1. Unless I missed it, the paper never mentions the evolution of the
   training error. For all I know by reading the paper and looking at
   the plots, the tested networks never overfit the training data, in
   which case I expect the training error to correlate to the
   validation set error even better than the proposed metric,
   something that would completely invalidate the claim of the paper
   (i.e., that the topological measure is telling us something
   nontrivial). In other words, in my opinion the interesting quantity
   to understand the learning dynamics is not the validation set error
   per se, but the generalization gap (or, in other words, the
   validation set error should be considered together with the
   training error, and not in isolation).
2. As far as I understand (although the paper is quite confusing in
   this respect - see below for more on this), the complex machinery
   of TDA is used to characterize how much the network changes during
   training. By looking at the plots in the paper, this quantity of
   interest (rate of change) is typically large at the beginning of
   the training, and small towards the end. Because this metric goes
   down quite quickly, its cumulative value during training takes on a
   saturating trend. This trend correlates with the validation set
   error, which also saturates. The main claim of the paper is that
   this correlation suggests a conceptual link between the two
   quantities. But there are many quantities that presumably exhibit
   similar trends during training! For instance, how about the norm of
   the weight update vector (which could be conceptualized as a much
   simpler way of measuring change in the network, and therefore could
   be a natural comparison to the proposed metric), or even just the
   learning rate, which is dynamically adjusted by RMSprop in the
   proposed case studies?  Similarly to my first point above, these
   are examples of quantities that could trivially correlate with the
   validation set error in the specific examples chosen, just as well
   or better than the proposed measure, without of course carrying any
   interesting information on the generalization gap.
3. Besides the issues above, the main quantitative piece of evidence
   proposed in the paper is the table with the correlation
   coefficients. To be honest, after reading the paper I am not sure
   what is being correlated with what here. On page 7 I read that "*For
   each experiment (e.g., layer size in MNIST), we plot both the
   evolution of the PH diagram distance and the validation score
   (accuracy). The plotted values are the corresponding means of the 5
   repetitions with different seeds. In addition, we compute the
   Pearson correlation for these values.*", and I deduce that the
   correlation must be between the validation score and the normalized
   cumulative topological distance (actually the plots contain also
   the non-cumulative distance, but I guess the correlation is
   computed with the cumulative one because the non-cumulative doesn't
   seem to be correlated with the validation score). However, later on
   in the same page I read "*...there is strong
   correlation. Intuitively, this is also observed in the plots,
   although once the distances are normalized it is not as clear to
   visualize.*". Does this mean that the correlation was computed on
   the unnormalized values, that is, not on what is in the plots,
   contradicting the statement above?
4. The paper does not seem to connect well with the existing
   theoretical literature on the generalization gap in deep
   learning. For instance, it completely ignores the effort to extend
   information-theoretic criteria to be applicable to deep
   networks. For instance, https://arxiv.org/abs/1906.07774v1 discuss
   how a technique introduced by Takeuchi in the '70s (itself a
   generalization of the Akaike Information Criterion) can be used to
   predict the generalization gap. The introduction of that paper also
   gives several references to multiple strands of the literature that
   seem relevant for anybody interested in proposing a new technique
   to estimate generalization (flatness etc). Other works that are
   relevant from a theoretical standpoint are recent approaches
   exploring an implicit simplicity bias in DNNs (see for instance
   https://arxiv.org/abs/1805.08522, https://arxiv.org/abs/1812.10156,
   and more recent literature citing those).

Minor points:
1. There are several passages in the paper that I completely failed to
   parse. For instance, "*Corneanu et al. (2020) try to estimate the
   performance gap between training and testing using PH
   measures. They claim. However, one can observe some caveats.*", "*We
   study the relation between the evolution of the PH diagram
   distances with the one of the validation score with the cumulative
   values of the distance between homologous persistence diagrams
   because this value seems much more stable. The information of the
   distance between the persistence diagrams has been normalized
   [...]*".
2. I don't understand why the y axes in most plots are labelled
   "distance difference" but the word "difference" never appears in
   the text of the paper - only distances are discussed.
3. The architecture and the pre-training of the convolutional network
   is entirely unclear to me. It is defined as "*In the case of CNNs,
   the pre-trained model is defined as 3 convolutional blocks with
   kernel size 3 (starting with 32 channels), interleaved with max
   pooling (its linear layers are thrown away after the
   pre-training)*". What does "starting with 32 channels" mean? This
   sounds like the authors imply that there is a standard way of
   changing the number of channels from one layer of a conv net to the
   next (this is not the case, as far as I know). How is max pooling
   performed? How many, and how large, linear layers are trained and
   then "thrown away" during pre-training? When does pre-training
   stop?
4. I do not agree with the following statement: "*If the measured
   distances are, indeed, related with the learning process of neural
   networks, these variations should not have any noticeable effect.*",
   when discussing the various network and training settings that were
   examined in the experiments. How is changing the learning rate
   expected not to have a noticeable effect on a metric related to the
   learning process? And more generally, why would one expect to
   change things like the size of the network in such a drastic way
   (going e.g. all the way down to 4 units per layer in size) and see
   no effect on the learning process?
5. When commenting on Comeanu et al 2020, the paper states that the
   method proposed there is "not usable in practice". I haven't read
   that paper, but I wonder to what degree this is just the opinion of
   the authors of the present work. If it is, it should be justified,
   and not passed on as a fact. Moreover, regardless of its merits (on
   the apparent lack of which I have commented in the Major Concerns
   section), the method proposed in the current work is enormously
   expensive from a computational standpoint (taking a week to run on
   a machine with 2 V100 and 1.5TB of RAM to analyze networks with a
   few hundred neurons on problems such as MNIST), so it doesn't seem
   like practical usability is a key area of focus of the authors.
6. The precise definition of the metric, and some of the choices
   behind it, are not clearly specified. On page 5, "*For each weighted
   directed graph associated with the state of a neural network, we
   link a directed flag complex to it. The topological properties of
   this directed flag complex are studied using homology groups H n
   . We calculate the homology groups up to degree 3 (H 0 -H 3 ).*"
   What is a flag complex? It is mentioned elsewhere in the paper that
   flag complexes are used in the other (unpublished, AFAICT) paper by
   the same authors, but the method proposed here should be
   understandable without having to go and read that other
   paper. Also, what motivates the choice of using homology groups up
   to degree 3?


**Summary Of The Paper:**

This paper presents some empirical observations about the relationship of a persistent homology-based measure of learning dynamics and validation set error, during the training of deep neural nets. The paper opens with an introduction on persistent homology, then introduces an approach to study the structure of deep nets using topological data analysis (TDA) tools. Three case studies are then presented, for which a measure of change in topological structure of the network during training is compared with the validation set error. The argument of the paper is that the two measures are correlated, and therefore it may be possible to use the topological measure (which depends on the structure of the network only, and not on data) in place of the validation set error to assess the generalization performance of a deep net.

**Summary Of The Review:**

Despite investigating the application of a promising set of techniques
to an interesting problem, this paper fails to sufficiently support
its claims. The main conclusion rests upon the correlation of a
proposed measure of change in deep nets during training with
validation set accuracy, but no effort has been made to control for
possible confounds that could explain such correlation
trivially. Moreover, multiple important passages in the paper are
extremely hard (or indeed impossible as far as I'm concerned) to
follow, and noticeable gaps are present in the references to related
literature.

---

> ### Author Response · Authors · 2021-11-21
> **Anwser 1 to Official Review of Paper4090 by Reviewer eLVu**
>
> >The argument of the paper is that the two measures are correlated, ...
>
> Topological information (persistente homology compared using persistence diagrams distances) anticipates validation accuracy of the model and only uses training information to anticipate validation accuracy.
>
> >Empirical studies can be valuable, but the complete absence of any theoretical justification ...
>
> The theoretical justification that could support this experimental paper would be a completely new theory of function approximation. This theory is not available and probably was not presented in this publication. Our paper is only an experimental study.
> We analyze how a numerical convergence, governed by the backpropagation method, is related to a topological convergence associated to the function but showing distinct properties of the convergence process, in our point of view, it is a parallel convergence process with interesting properties: the anticipation of the validation accuracy using only training data. We understand that this is a relevant result for the study of generalization in NNs.
>
> >Unless I missed it, the paper never mentions the evolution of the training error. ...
>
> This work is focused on the analysis of the problem of the generalization of the NN, for this reason the validation accuracy is our main object of study. We consider that this proposal may be interesting to complement the current analysis.
>
> >As far as I understand (although the paper is quite confusing in this respect ...
> There are two points to note: first, this two convergence processes does not occur at the same time. The topological convergence seems to anticipate the convergence of the validation accuracy. The topological convergence is calculated on training data that exclude the validation set. This is not the case for the validation error which is calculated on the validation set.
>
> >The main claim of the paper is that this correlation suggests a conceptual link between the two quantities. ...
>
> Do any of these quantities (norm of the weight update, learning rate) anticipate validation accuracy using training data only? Are there any references in the literature to support this?
>
> >Besides the issues above, the main quantitative piece of evidence proposed in the paper ...
> The validation error is calculated at the end of each epoch while the distances between persistence diagrams are calculated after each batch. As can be seen in Figure 6 (the figure can be enlarged), for each batch, the changes in the distance of the persistence diagrams are not smooth (mainly because we are studying discrete homology groups using, in turn, a discretization of the persistence diagrams themselves).
>
> >However, later on in the same page I read "...there is strong correlation...
>
> As can be read in the title of the correlation table 1, the correlations have been calculated at the 20 points corresponding to the epochs (where we have validation data) between the validation values and the accumulated topological distance.
>
> >The paper does not seem to connect well with the existing theoretical literature on the generalization gap ...
>
> Thank you for the references. We will study them and analyze their relevance to this study.
>
> >There are several passages in the paper that I completely failed to parse. ...
>
> Ok. We will review them
>
> >I don't understand why the y axes in most plots are labelled "distance difference" ...
>
> Ok. The correct plot title is PD distance.
>
> >The architecture and the pre-training of the convolutional network is entirely unclear to me. ...
>
> 32 channels, not 32 filters.
>
> >I do not agree with the following statement: "If the measured distances are, indeed, ...
>
> As a control experiment, for each training configuration, we performed 5 random initializations of the weights and 5 rearrangements of the training examples. Of course the change in learning rate produces changes in training results as shown in Figure 9.
>
> >And more generally, why would one expect to change ...
>
> We study the learning process in different learning scenarios. These scenarios include significant modifications in the architecture.
>
> >When commenting on Comeanu et al 2020, ...
>
> We do not use the representation from Coneanu as activations are associated with concrete input data. We are interested into study NN structure as a function not as function evaluated on specific data.
> We have questions regarding the use of activations:
>
> - How much data do you pass through the network?
> - It is not the same to pass {training, validation, testing, production} data. Isn't it?
> - The subset of data selected may not be representative.
> - Information flow of the network (weights) might not be taken into account completely.
> In our study,  activation information is not usable in practice because validation data is not used during training process.

---

> > ### Comment · Reviewer_eLVu · 2021-11-21
> > **[part 1] This answer is extremely hard to read; and from what I can understand, it fails to address my concerns**
> >
> > First off, this answer is extremely hard to read and follow.
> >  - I am quoted writing things that I didn't. Some of it looks like a
> >    series of formatting errors, where the authors inserted their own
> >    answers inside the quotations from my review. Some of it looks like
> >    the authors are mixing up the different reviews (for instance, I
> >    think it was another reviewer that raised the point about
> >    validation error being computed at the end of each epoch while
> >    persistence diagrams are computed for each batch). As a tip for
> >    future paper submission, the authors should make sure they
> >    understand how Markdown works, and that they do not mix up distinct
> >    reviewers in their rebuttal.
> >  - The liberal use of ellipsis to shorten the quotation from my review
> >    make it harder than necessary to understand the link between the
> >    points of my review and the corresponding answer by the
> >    authors. I have no idea what possible advantage could there be
> >    in doing this.
> >
> > More generally, many of the responses seem entirely disconnected from
> > the criticism that they are supposed to respond to.Here are a few examples.
> >
> > 1. In my major point 3 I wrote:
> >
> > >Besides the issues above, the main quantitative piece of evidence
> > proposed in the paper is the table with the correlation
> > coefficients. To be honest, after reading the paper I am not sure what
> > is being correlated with what here. On page 7 I read that "For each
> > experiment (e.g., layer size in MNIST), we plot both the evolution of
> > the PH diagram distance and the validation score (accuracy). The
> > plotted values are the corresponding means of the 5 repetitions with
> > different seeds. In addition, we compute the Pearson correlation for
> > these values.", and I deduce that the correlation must be between the
> > validation score and the normalized cumulative topological distance
> > (actually the plots contain also the non-cumulative distance, but I
> > guess the correlation is computed with the cumulative one because the
> > non-cumulative doesn't seem to be correlated with the validation
> > score). However, later on in the same page I read "...there is strong
> > correlation. Intuitively, this is also observed in the plots, although
> > once the distances are normalized it is not as clear to
> > visualize.". Does this mean that the correlation was computed on the
> > unnormalized values, that is, not on what is in the plots,
> > contradicting the statement above?
> >
> > To which the authors replied:
> >
> > >The validation error is calculated at the end of each epoch while the
> > distances between persistence diagrams are calculated after each
> > batch. As can be seen in Figure 6 (the figure can be enlarged), for
> > each batch, the changes in the distance of the persistence diagrams
> > are not smooth (mainly because we are studying discrete homology
> > groups using, in turn, a discretization of the persistence diagrams
> > themselves).
> >
> > I have no idea what this means in this context. It seems like a
> > copy-paste from an answer to a different point raised by another
> > reviewer.
> >
> > 2. In my minor point 3, I wrote:
> >
> > >The architecture and the pre-training of the convolutional network is
> >  entirely unclear to me. It is defined as "In the case of CNNs, the
> >  pre-trained model is defined as 3 convolutional blocks with kernel
> >  size 3 (starting with 32 channels), interleaved with max pooling (its
> >  linear layers are thrown away after the pre-training)". What does
> >  "starting with 32 channels" mean? This sounds like the authors imply
> >  that there is a standard way of changing the number of channels from
> >  one layer of a conv net to the next (this is not the case, as far as
> >  I know). How is max pooling performed? How many, and how large,
> >  linear layers are trained and then "thrown away" during pre-training?
> >  When does pre-training stop?
> >
> > In their response, the authors shortened this to
> > >The architecture and
> > the pre-training of the convolutional network is entirely unclear to
> > me. ...
> >
> > and answered:
> >
> > >32 channels, not 32 filters.
> >
> > How this is an
> > answer to my point, which included four explicit questions (just
> > counting by the number of question marks), is beyond me.

---

> > > ### Comment · Reviewer_eLVu · 2021-11-21
> > > **[part 2]**
> > >
> > > 3. Similarly, in my minor concern #5, I state that if the authors want to
> > > dismiss the method in Comeanu 2020 as "not usable in practice",
> > >    they should provide some evidence to back up that statement (*I
> > >    wonder to what degree this is just the opinion of the authors of
> > >    the present work. If it is, it should be justified, and not passed
> > >    on as a fact*). To this observation, I got the following answer:
> > >
> > > >We do not use the representation from Coneanu as activations are
> > > associated with concrete input data. We are interested into study NN
> > > structure as a function not as function evaluated on specific data.
> > > We have questions regarding the use of activations:
> > > > - How much data do you pass through the network?
> > > > - It is not the same to pass {training, validation, testing, production} data. Isn't it?
> > > > - The subset of data selected may not be representative.
> > > > - Information flow of the network (weights) might not be taken into
> > >   account completely. In our study, activation information is not
> > >   usable in practice because validation data is not used during
> > >   training process.
> > >
> > > I frankly do not have the slightest clue as to how this is relevant to
> > > my point. And if I may offer a word of advice for future submissions:
> > > passive-aggressive rhetorical questions are typically not very
> > > effective at winning over your interlocutor (but again, in this case
> > > the issue is simply that I have no idea how any of this relates to
> > > what I asked).
> > >
> > > ---
> > >
> > > Overall, it seems that the authors decided not to make a substantive
> > > effort to address my points, so I will not invest more of my time
> > > explaining why answers like the one I just cited do not advance the
> > > case for the acceptance of this paper.
> > >
> > > However, in the interest of improving this paper for a future
> > > submission, I will now elaborate more on why this response does not
> > > address a fundamental criticism that appeared both in my review and as
> > > the main concern of Reviewer JhC7 (who, I feel, has expressed this
> > > idea better than I did). To present the argument in its simplest
> > > possible form, it is as follows: the fact that the homology-based
> > > measure of network change is correlated with the validation set error
> > > is not enough to support the claim that the topological measure does
> > > anything nontrivial, or that it would still continue to predict the
> > > validation set error even in an overfitting regime. The observed
> > > correlation is insufficient evidence because, as both myself and JhC7
> > > have noted, for all we know, the networks in the paper are never
> > > overfitting. Therefore, we would expect the validation set error to
> > > correlate very well with the training set error (as well as other
> > > metrics, unrelated to generalization). To this point, the authors
> > > reply in several, scattered places.
> > >
> > > > From our point of view, the possible confounding factors have been
> > > studied (dropout, learning rate, architectures, problems). In
> > > addition, control experiments have been included (order examples and
> > > initializations of the weights).
> > >
> > > This just amount to saying, with no justification, that the confound I
> > > was talking about is not one of "the possible confounding
> > > factors". Needless to say, I'm hardly satisfied by this response.
> > >
> > > >This work is focused on the analysis of the problem of the
> > > generalization of the NN, for this reason the validation accuracy is
> > > our main object of study. We consider that this proposal may be
> > > interesting to complement the current analysis.
> > >
> > > This is another instance of non-sequitur in the rebuttal, made worse
> > > by the selective quoting used by the authors, which (as for most of
> > > their replies) omitted the main point in the part of the review that
> > > they were responding to. This was ostensibly written in answer to the
> > > following (major concern 1): "*Unless I missed it, the paper never
> > > mentions the evolution of the training error. For all I know by
> > > reading the paper and looking at the plots, the tested networks never
> > > overfit the training data, in which case I expect the training error
> > > to correlate to the validation set error even better than the proposed
> > > metric, something that would completely invalidate the claim of the
> > > paper (i.e., that the topological measure is telling us something
> > > nontrivial).* (etc)". So, in other words, I raised the concern that the
> > > proposed metric may be trivial because it could simply reflect the
> > > correlation of the training error with the validation error, and the
> > > authors' response was that the validation accuracy is "the main object
> > > of study". Again, how is this an answer? If the validation error and
> > > the training error are strongly correlated, how can you distinguish
> > > between whether your measure correlates with one or the other?

---

> > > > ### Comment · Reviewer_eLVu · 2021-11-21
> > > > **[part 3]**
> > > >
> > > > ## Conclusion
> > > > In conclusion, the authors' response was extremely hard to read,
> > > > misquoted my review, provided several rebuttal points that I could only describe as non-sequiturs, and refused to acknowledge that any change
> > > > could be made to the paper (apart from some generic "we will review"
> > > > statements about missing acknowledgements to important, existing
> > > > literature, or to some unclear passages).
> > > >
> > > > I will not be upgrading my score.

---

> > > > > ### Comment · Reviewer_Fo4q · 2021-11-21
> > > > > **Thank you for your candour!**
> > > > >
> > > > > As a fellow reviewer on this paper whose review unfortunately received no response thus far, I want to point out that I acknowledge the candour of your comments here.
> > > > >
> > > > > I would very much like help the authors to improve their paper. The responses I have seen here so far are not encouraging. In case the authors are reading this, I would really like to urge them to treat this feedback seriously—from what I see here is that all reviewers almost unanimously pointed out the _same_ issues with the current version of the paper.
> > > > >
> > > > > I do believe in the potential of the idea; at the same time, I also see that some additional updates are to be performed. I understand that this is not the desired outcome, yet I want to stress that we are all on the same page here and want to contribute to having a conference with great papers that will advance our field.

---

> > > > > > ### Comment · Reviewer_eLVu · 2021-11-21
> > > > > > **Thanks**
> > > > > >
> > > > > > Thank you! You're right, we're all on the same team and our common goal is a great conference with solid papers that advance the field.
> > > > > >
> > > > > > In my opinion, this means that as reviewers we should be generous with our time and effort. We are all busy and pressed for time, and there is no immediate reward for a thoughtful review. It is easy to write vague or subjective comments, either supporting or opposing a paper. It takes more work to be specific and constructive, to raise questions that could be addressed in concrete ways, for the benefit of the work being discussed.
> > > > > >
> > > > > > As authors, as you say, we should take seriously the feedback we get. Of course there will be misunderstandings and "noise" of various kinds (I know that as a reviewer I have made mistakes in the past, despite my best efforts, and I certainly will make more in the future), but we should be on the lookout for the signal amidst this noise. This signal is, in a way, free work done by the reviewers for us.
> > > > > >
> > > > > > So to bring this back to the current paper: as you say, this is work that has potential, but when multiple independent reviews agree on asking the same questions/raising the same issues, chances are that these issues are worth one's attention.

---

> ### Author Response · Authors · 2021-11-21
> **Anwser 2 to Official Review of Paper4090 by Reviewer eLVu**
>
> >the method proposed in the current work is enormously expensive from a computational standpoint (taking a week to run on a machine with 2 V100 and 1.5TB of RAM to analyze networks with a few hundred neurons on problems such as MNIST), so it doesn't seem like practical usability is a key area of focus of the authors.
>
> The experiments performed are computationally expensive due to the large number of combinations of experiments performed (dropout, learning rate, architectures, problems, weight initialization, training example errors, ...).
> There are many ways to speed up the proposed computations (precision of the persistent homology parameter, vectorization dimension, etc.). In this paper, we have tried to carry out the study with as much detailed precision as possible.
>
> >The precise definition of the metric, and some of the choices behind it, are not clearly specified. On page 5, "For each weighted directed graph associated with the state of a neural network, we link a directed flag complex to it. The topological properties of this directed flag complex are studied using homology groups H n . We calculate the homology groups up to degree 3 (H 0 -H 3 )." What is a flag complex?
> It is mentioned elsewhere in the paper that flag complexes are used in the other (unpublished, AFAICT) paper by the same authors, but the method proposed here should be understandable without having to go and read that other paper.
> Flag complex definition is included in the supplementary material (Definition 3).
>
> >Also, what motivates the choice of using homology groups up to degree 3?
> In the study we were interested in the analysis of high degree homology groups. The motivation for this choice H_3 is due to our computational limitation. As mentioned above, given the large number of combinations contained in the present paper, the maximum degree of homology groups has been limited to 3. In practical applications this number can be increased.
>
> >Despite investigating the application of a promising set of techniques to an interesting problem, this paper fails to sufficiently support its claims. The main conclusion rests upon the correlation of a proposed measure of change in deep nets during training with validation set >
> accuracy, but no effort has been made to control for possible confounds that could explain such correlation trivially.
>
> From our point of view, the possible confounding factors have been studied (dropout, learning rate, architectures, problems). In addition, control experiments have been included (order examples and initializations of the weights).
>
> >Moreover, multiple important passages in the paper are extremely hard (or indeed impossible as far as I'm concerned) to follow
>
> Ok. We will review them
>
> >noticeable gaps are present in the references to related literature.
>
> Ok. We will review them

---

> ### Author Response · Authors · 2021-11-22
> **Anwser 1 to Official Review of Paper4090 by Reviewer eLVu**
>
> >The argument of the paper is that the two measures are correlated, and therefore it may be possible to use the topological measure (which depends on the structure of the network only, and not on data) in place of the validation set error to assess the generalization performance of a deep net.
>
> Topological information (persistente homology compared using persistence diagrams distances) anticipates validation accuracy of the model and only uses training information to anticipate validation accuracy.
>
> >Empirical studies can be valuable, but the complete absence of any theoretical justification (or even just an intuitively plausible story) for the main argument means that the experimental evidence presented carries all the burden of convincing the reader. My main concern with the paper is that, unfortunately, the analyses presented are nowhere close to the standard of rigor and thoroughness one would expect in this case. In particular:
>
> The theoretical justification that could support this experimental paper would be a completely new theory of function approximation. This theory is not available and probably was not presented in this publication. Our paper is only an experimental study.
> We analyze how a numerical convergence, governed by the backpropagation method, is related to a topological convergence associated to the function but showing distinct properties of the convergence process, in our point of view, it is a parallel convergence process with interesting properties: the anticipation of the validation accuracy using only training data. We understand that this is a relevant result for the study of generalization in NNs.
>
> >Unless I missed it, the paper never mentions the evolution of the training error. For all I know by reading the paper and looking at the plots, the tested networks never overfit the training data, in which case I expect the training error to correlate to the validation set error even better than the proposed metric, something that would completely invalidate the claim of the paper (i.e., that the topological measure is telling us something nontrivial). In other words, in my opinion the interesting quantity to understand the learning dynamics is not the validation set error per se, but the generalization gap (or, in other words, the validation set error should be considered together with the training error, and not in isolation).
>
> This work is focused on the analysis of the problem of the generalization of the NN, for this reason the validation accuracy is our main object of study. We consider that this proposal may be interesting to complement the current analysis.
>
> >As far as I understand (although the paper is quite confusing in this respect - see below for more on this), the complex machinery of TDA is used to characterize how much the network changes during training. By looking at the plots in the paper, this quantity of interest (rate of change) is typically large at the beginning of the training, and small towards the end. Because this metric goes down quite quickly, its cumulative value during training takes on a saturating trend. This trend correlates with the validation set error, which also saturates.
>
> There are two points to note: first, this two convergence processes does not occur at the same time. The topological convergence seems to anticipate the convergence of the validation accuracy. The topological convergence is calculated on training data that exclude the validation set. This is not the case for the validation error which is calculated on the validation set.
>
> >The main claim of the paper is that this correlation suggests a conceptual link between the two quantities. But there are many quantities that presumably exhibit similar trends during training! For instance, how about the norm of the weight update vector (which could be conceptualized as a much simpler way of measuring change in the network, and therefore could be a natural comparison to the proposed metric), or even just the learning rate, which is dynamically adjusted by RMSprop in the proposed case studies? Similarly to my first point above, these are examples of quantities that could trivially correlate with the validation set error in the specific examples chosen, just as well or better than the proposed measure, without of course carrying any interesting information on the generalization gap.
>
> Do any of these quantities (norm of the weight update, learning rate) anticipate validation accuracy using training data only? Are there any references in the literature to support this?

---

> ### Author Response · Authors · 2021-11-22
> **Anwser 2 to Official Review of Paper4090 by Reviewer eLVu**
>
> >Besides the issues above, the main quantitative piece of evidence proposed in the paper is the table with the correlation coefficients. To be honest, after reading the paper I am not sure what is being correlated with what here. On page 7 I read that "For each experiment (e.g., layer size in MNIST), we plot both the evolution of the PH diagram distance and the validation score (accuracy). The plotted values are the corresponding means of the 5 repetitions with different seeds. In addition, we compute the Pearson correlation for these values.", and I deduce that the correlation must be between the validation score and the normalized cumulative topological distance (actually the plots contain also the non-cumulative distance, but I guess the correlation is computed with the cumulative one because the non-cumulative doesn't seem to be correlated with the validation score).
>
> The validation error is calculated at the end of each epoch while the distances between persistence diagrams are calculated after each batch. As can be seen in Figure 6 (the figure can be enlarged), for each batch, the changes in the distance of the persistence diagrams are not smooth (mainly because we are studying discrete homology groups using, in turn, a discretization of the persistence diagrams themselves).
>
> >However, later on in the same page I read "...there is strong correlation. Intuitively, this is also observed in the plots, although once the distances are normalized it is not as clear to visualize.". Does this mean that the correlation was computed on the unnormalized values, that is, not on what is in the plots, contradicting the statement above?
>
> As can be read in the title of the correlation table 1, the correlations have been calculated at the 20 points corresponding to the epochs (where we have validation data) between the validation values and the accumulated topological distance.
>
> >The paper does not seem to connect well with the existing theoretical literature on the generalization gap in deep learning. For instance, it completely ignores the effort to extend information-theoretic criteria to be applicable to deep networks. For instance, https://arxiv.org/abs/1906.07774v1 discuss how a technique introduced by Takeuchi in the '70s (itself a generalization of the Akaike Information Criterion) can be used to predict the generalization gap. The introduction of that paper also gives several references to multiple strands of the literature that seem relevant for anybody interested in proposing a new technique to estimate generalization (flatness etc). Other works that are relevant from a theoretical standpoint are recent approaches exploring an implicit simplicity bias in DNNs (see for instance https://arxiv.org/abs/1805.08522, https://arxiv.org/abs/1812.10156, and more recent literature citing those).
>
> Thank you for the references. We will study them and analyze their relevance to this study.
>
> >There are several passages in the paper that I completely failed to parse. For instance, "Corneanu et al. (2020) try to estimate the performance gap between training and testing using PH measures. They claim. However, one can observe some caveats.", "We study the relation between the evolution of the PH diagram distances with the one of the validation score with the cumulative values of the distance between homologous persistence diagrams because this value seems much more stable. The information of the distance between the persistence diagrams has been normalized [...]".
>
> Ok. We will review them
>
> >I don't understand why the y axes in most plots are labelled "distance difference" but the word "difference" never appears in the text of the paper - only distances are discussed.
>
> Ok. The correct plot title is PD distance.
>
> >The architecture and the pre-training of the convolutional network is entirely unclear to me. It is defined as "In the case of CNNs, the pre-trained model is defined as 3 convolutional blocks with kernel size 3 (starting with 32 channels), interleaved with max pooling (its linear layers are thrown away after the pre-training)". What does "starting with 32 channels" mean? This sounds like the authors imply that there is a standard way of changing the number of channels from one layer of a conv net to the next (this is not the case, as far as I know). How is max pooling performed? How many, and how large, linear layers are trained and then "thrown away" during pre-training? When does pre-training stop?
>
> This is indeed an error. There are 32 filters.

---

> ### Author Response · Authors · 2021-11-22
> **Anwser 3 to Official Review of Paper4090 by Reviewer eLVu**
>
> >I do not agree with the following statement: "If the measured distances are, indeed, related with the learning process of neural networks, these variations should not have any noticeable effect.", when discussing the various network and training settings that were examined in the experiments. How is changing the learning rate expected not to have a noticeable effect on a metric related to the learning process?
>
> As a control experiment, for each training configuration, we performed 5 random initializations of the weights and 5 rearrangements of the training examples. Of course the change in learning rate produces changes in training results as shown in Figure 9.
>
> >And more generally, why would one expect to change things like the size of the network in such a drastic way (going e.g. all the way down to 4 units per layer in size) and see no effect on the learning process?
>
> We study the learning process in different learning scenarios. These scenarios include significant modifications in the architecture.
>
> >When commenting on Corneanu et al 2020, the paper states that the method proposed there is "not usable in practice". I haven't read that paper, but I wonder to what degree this is just the opinion of the authors of the present work. If it is, it should be justified, and not passed on as a fact. Moreover, regardless of its merits (on the apparent lack of which I have commented in the Major Concerns section)
>
> We do not use the representation from Corneanu as activations are associated with concrete input data. We are interested into study NN structure as a function not as function evaluated on specific data.
>
> We have questions regarding the use of activations:
> - How much data do you pass through the network?
> - It is not the same to pass {training, validation, testing, production} data. Isn't it?
> - The subset of data selected may not be representative.
> - Information flow of the network (weights) might not be taken into account completely.
> In our study,  activation information is not usable in practice because validation data is not used during training process.
>
> >the method proposed in the current work is enormously expensive from a computational standpoint (taking a week to run on a machine with 2 V100 and 1.5TB of RAM to analyze networks with a few hundred neurons on problems such as MNIST), so it doesn't seem like practical usability is a key area of focus of the authors.
>
> The experiments performed are computationally expensive due to the large number of combinations of experiments performed (dropout, learning rate, architectures, problems, weight initialization, training example errors, ...).
> There are many ways to speed up the proposed computations (precision of the persistent homology parameter, vectorization dimension, etc.). In this paper, we have tried to carry out the study with as much detailed precision as possible.
>
> >The precise definition of the metric, and some of the choices behind it, are not clearly specified. On page 5, "For each weighted directed graph associated with the state of a neural network, we link a directed flag complex to it. The topological properties of this directed flag complex are studied using homology groups H n . We calculate the homology groups up to degree 3 (H 0 -H 3 )." What is a flag complex?
> It is mentioned elsewhere in the paper that flag complexes are used in the other (unpublished, AFAICT) paper by the same authors, but the method proposed here should be understandable without having to go and read that other paper.
> Flag complex definition is included in the supplementary material (Definition 3).
>
> >Also, what motivates the choice of using homology groups up to degree 3?
> In the study we were interested in the analysis of high degree homology groups. The motivation for this choice H_3 is due to our computational limitation. As mentioned above, given the large number of combinations contained in the present paper, the maximum degree of homology groups has been limited to 3. In practical applications this number can be increased.
>
> >Despite investigating the application of a promising set of techniques to an interesting problem, this paper fails to sufficiently support its claims. The main conclusion rests upon the correlation of a proposed measure of change in deep nets during training with validation set >
> accuracy, but no effort has been made to control for possible confounds that could explain such correlation trivially.
>
> From our point of view, the possible confounding factors have been studied (dropout, learning rate, architectures, problems). In addition, control experiments have been included (order examples and initializations of the weights).
>
> >Moreover, multiple important passages in the paper are extremely hard (or indeed impossible as far as I'm concerned) to follow
>
> Ok. We will review them
>
> >noticeable gaps are present in the references to related literature.
>
> Ok. We will review them

---

### Official Review · Reviewer_nK7m · 2021-11-04

**Correctness:** 1
**Technical Novelty And Significance:** 3
**Empirical Novelty And Significance:** 3
**Recommendation:** 3
**Confidence:** 4

**Main Review:**

I think the subject of the paper is interesting itself and using PH techniques to discover insights about deep learning models is indeed an interesting topic. However, I find the paper to be still in early stage of development and still requires work. Specifically, It think the authors need to really support  their claim for the title. Generalization is probably the hardest problem in deep learning model models and the claim that PH truly captures that is not explained well at least at the beginning of the paper. What is the metric for such a claim ? The authors mentioned "that there exists a high correlation with the corresponding validation accuracy of the model." I do not think think that is enough reason to justify the claim of the paper.

The contributions are not very novel in my opinion. For instance :

Based on principles of Algebraic Topology, we propose measuring the distances (Silhouette
and Heat) between the PH persistence diagrams obtained from a given state of a neural
network during the training procedure and the one in the immediately previous weights
update.

Measuring the distance between two weights of neural network is not significant contribution, unless you do something with it--which you are claiming next but this particular point is not a valid contribution in my opinion.

**Summary Of The Paper:**

The paper uses distances between the Persistent diagrams of states of neural networks and observe that during training there is a high correlation with the corresponding validation accuracy of the model. The authors tested their method on a variety of datasets.

**Summary Of The Review:**

I am sorry to say that I have to reject the paper. The paper is still in its early stage and requires more careful presentation especially in the beginning to make the contribution more clear and crisp

---

### Author Response · Authors · 2021-11-21
**Answer to reviewer nK7m**

Thanks for your comments.

> I think the subject of the paper is interesting itself and using PH techniques to discover insights about deep learning models is indeed an interesting topic. However, I find the paper to be still in early stage of development and still requires work. Specifically, It think the authors need to really support their claim for the title. Generalization is probably the hardest problem in deep learning model models and the claim that PH truly captures that is not explained well at least at the beginning of the paper. What is the metric for such a claim ? The authors mentioned "that there exists a high correlation with the corresponding validation accuracy of the model." I do not think think that is enough reason to justify the claim of the paper

Validation accuracy is understood as a proxy of the generalization error across the machine learning community. Our proposed distance strongly correlates with validation accuracy. We don't claim to have solved the theoretical foundations of the generalization of neural networks whatsoever.

> The contributions are not very novel in my opinion. For instance :
Based on principles of Algebraic Topology, we propose measuring the distances (Silhouette and Heat) between the PH persistence diagrams obtained from a given state of a neural network during the training procedure and the one in the immediately previous weights update.
Measuring the distance between two weights of neural network is not significant contribution,

We don't measure the distance between "two weights of neural networks". We characterize neural networks as graphs, compute topological distances between them and find that this distance strongly correlates with validation accuracy. If you say it's not novel, can you point to any particular previous work doing anything remotely similar to this?

> unless you do something with it

Yes, we do. We find that this distance strongly correlates with validation accuracy and thus could potentially be used for the exact same things validation accuracy is currently used.

> --which you are claiming next but this particular point is not a valid contribution in my opinion.

Please, could you develop? Why?

---

### Comment · Reviewer_eLVu · 2021-11-22
**It is very bad form to delete your messages after others have already responded to them**

For anybody who may look at the history of the discussion (area chairs etc): the authors have deleted their response to my review, **after** I had already replied at length to it. They then reposted their response as a separate set of messages, with minor formatting and language tweaks (in particular, this was **not** done in order to formulate a better response to my initial review that engaged also with my later comments; there are no substantial differences from what the authors wrote initially). This makes it very hard to trace the flow of the conversation, and strips my own messages of their context.

Needless to say, I'm not going back and moving my own responses to these new threads.

---

> ### Author Response · Authors · 2021-11-22
> **Reply to It is very bad form to delete your messages after others have already responded to them**
>
> Excuse me, didn't you tell me in your message that you couldn't understand my answers because you couldn't find the complete quote?
>
> The reason I had not included, on the deleted message, the full quote from your reply was because it did not fit in the space provided by OpenReview message box. I created multiple replies by chunking the original message with the same content.
>
> I do not want you to interpret the republishing of the message in that way.

---

> > ### Comment · Reviewer_eLVu · 2021-11-22
> > **Thank you for restoring the text of your original responses**
> >
> > I see that you have restored your initial responses in the messages I replied to. Thank you, that's better, as now the context of the discussion is preserved.

---

### Decision · Program_Chairs · 2022-01-20

**Decision:**

Reject

**Comment:**

This paper got uniformly strongly negative reviews.  The issue of estimating or bounding generalization accuracy from performance on the training set has a huge history and literature.  After considerable discussion the reviewers uniformly find this paper lacking in making a contribution to that literature.